# Transcriptomic Analysis Reveals the High-Oleic Acid Feedback Regulating the Homologous Gene Expression of Stearoyl-ACP Desaturase 2 (*SAD2*) in Peanuts

**DOI:** 10.3390/ijms20123091

**Published:** 2019-06-25

**Authors:** Hao Liu, Jianzhong Gu, Qing Lu, Haifen Li, Yanbin Hong, Xiaoping Chen, Li Ren, Li Deng, Xuanqiang Liang

**Affiliations:** 1Crops Research Institute, Guangdong Academy of Agricultural Sciences, South China Peanut Sub-Center of National Center of Oilseed Crops Improvement, Guangdong Key Laboratory of Crop Genetic Improvement, Guangzhou 510640, China; liuhao2054@stu.scau.edu.cn (H.L.); luqing@gdaas.cn (Q.L.); lihaifen@gdaas.cn (H.L.); hongyanbin@gdaas.cn (Y.H.); chenxiaoping@gdaas.cn (X.C.); 2Peanut Research Institute, Kaifeng Academy of Agriculture and Forestry, Kaifeng 475004, China; xinkeyan@126.com (J.G.); renli120@sina.com (L.R.); dengli_1225@sina.com (L.D.)

**Keywords:** peanut, *FAD2*, transcriptome, oleic acid, *SAD2*, fatty acid desaturase

## Abstract

Peanuts with high oleic acid content are usually considered to be beneficial for human health and edible oil storage. In breeding practice, peanut lines with high monounsaturated fatty acids are selected using *fatty acid desaturase 2* (*FAD2*), which is responsible for the conversion of oleic acid (C18:1) to linoleic acid (C18:2). Here, comparative transcriptomics were used to analyze the global gene expression profile of high- and normal-oleic peanut cultivars at six time points during seed development. First, the mutant type of *FAD2* was determined in the high-oleic peanut (H176). The result suggested that early translation termination occurred simultaneously in the coding sequence of *FAD2-A* and *FAD2-B*, and the cultivar H176 is capable of utilizing a potential germplasm resource for future high-oleic peanut breeding. Furthermore, transcriptomic analysis identified 74 differentially expressed genes (DEGs) involved in lipid metabolism in high-oleic peanut seed, of which five DEGs encoded the *fatty acid desaturase*. *Aradu.XM2MR* belonged to the homologous gene of *stearoyl-ACP (acyl carrier protein*) *desaturase 2* (*SAD2*) that converted the C18:0 into C18:1. Further subcellular localization studies indicated that *FAD2* was located at the endoplasmic reticulum (ER), and *Aradu.XM2MR* was targeted to the plastid in *Arabidopsis* protoplast cells. To examine the dynamic mechanism of this finding, we focused on the peroxidase (POD)-mediated fatty acid (FA) degradation pathway. The *fad2* mutant significantly increased the POD activity and H_2_O_2_ concentration at the early stage of seed development, implying that redox signaling likely acted as a messenger to connect the signaling transduction between the high-oleic content and *Aradu.XM2MR* transcription level. Taken together, transcriptome analysis revealed the feedback mechanism of *SAD2* (*Aradu.XM2MR*) associated with *FAD2* mutation during the seed developmental stage, which could provide a potential peanut breeding strategy based on identified candidate genes to improve the content of oleic acid.

## 1. Introduction

The peanut (*Arachis hypogaea* L.) is a vital mercantile crop and is widely cultivated in tropical and sub-tropical regions, with a global production of more than 40 million tons per year. It provides stable nourishment for many human populations [1]. Peanut kernels are rich in natural nutrients such as proteins, fatty acids (FAs), vitamins, minerals, and fibers. Moreover, edible oil is extracted from eleven million tons of peanut seeds in the annual world food industry [2]. Peanut oil contains approximately 80% unsaturated fatty acids (USFAs) and fewer than 15% saturated fats. USFAs are mostly composed of 50% monounsaturated FA oleic acid (OA, C18:1) and 30% polyunsaturated FA linoleic acid (LOA, C18:2) [3]. Edible peanut oils with high levels of OA have been confirmed to have positive effects on human health, including decreasing the risk of coronary heart diseases and decreasing cholesterol levels [4]. Therefore, with the increased attention on the health benefits of OA, there is a need to develop a market-oriented peanut cultivar with high levels of OA [5]. 

OA is classified as a monounsaturated omega-9 fatty acid with the formula of C_18_H_34_O_2_, and the double bond is located at position δ9 [6]. The majority of vegetable OA is commonly present in the form of monoglycerides, diglycerides, and triglycerides. The *de novo* synthesis of OA in plants occurs in the plastid with initial carbon flux from pyruvate to acetyl-CoA, which is subsequently used for the synthesis of palmitoyl-ACP (C16:0-ACP) by *3-ketoacyl-ACP synthase* (*KAS*) *III* and *KAS I*. *KAS II* catalyzes the further conversion of C16:0-ACP to stearoyl-ACP (C18:0-ACP) [7]. The biosynthesis of OA continues with the dehydrogenation of stearoyl-ACP to oleoyl-ACP (C18:1-ACP), catalyzed by s*tearoyl-CoA 9-desaturase 2* (*SAD2*). In fact, oleoyl-ACP is exported from the chloroplast into the endoplasmic reticulum to promote OA synthesis by the acetyl-CoA transport pathway [8]. In addition, OA can be converted into LOA by the crucial enzyme *fatty acid desaturase 2* (*FAD2*), which catalyzes the dehydrogenation at the δ12 position of the hydrocarbon chain. Peanuts with high OA are usually obtained from *FAD2* mutants [9]. Currently, *AhFAD2A* [10] and *AhFAD2B* [10] are cloned from diploid ancestors A and B’s subgenomes, respectively. Both of them encode δ12 FA desaturase to predominantly modulate the critical conversion of OA to LOA. Moreover, the recessive mutant allele *ahfad2* also induces OA accumulation [11]. To date, six candidate isoforms of *AhFAD2* have been identified, which can facilitate further genetic manipulation of peanut oil quality [12]. Nonetheless, a comprehensive illustration of the OA formation pathways and the molecular functions of the homologous gene *AhFAD2* in peanut remains to be accomplished. 

The modern peanut is an allotetraploid crop that is derived from multiple hybridization events between diploid ancestor *Arachis ipaensis* (BB) and diverse *Arachis duranensis* (AA) during its origination on the South American continent [13]. The peanut genome is approximately 2.7 GB in size, and the complete nucleotide sequence was recently released [14,15,16]. However, it is difficult to assemble the cultivated peanut genome due to the close genetic relationship between the two diploid species [17]. One strategy is to obtain the whole genome sequences of *Arachis duranensis* and *Arachis ipaensis* and use them as the reference to assemble the tetraploid genome [18]. Without the genome sequence of the cultivated peanut, assembled transcripts based on the diploid ancestor’s subgenome are a complimentary resource for establishing a global gene atlas of *Arachis hypogaea*. To date, transcriptome-based high-throughput sequencing has generated large volumes of data associated with lipid metabolism in the peanut [19]. For instance, *de novo* transcriptome analysis not only reveals the expression patterns of lipid genes during the development of peanut seeds, but also sheds light on the molecular mechanisms of lipid metabolism and highlights rate-limiting enzymes for lipid storage in peanuts with different oil contents [20]. Furthermore, the genetically controlled pod-filling stage is a key point for the development of peanut seeds. Identification of differentially expressed genes involved in peanut oil synthesis would explain the genotypic variation in oil biosynthesis and pod-filling, and it can provide the genetic basis for understanding seed development and oil accumulation in the peanut [21]. 

Previous studies focused mainly on the identification of various alleles of *fad2* in high-oleate peanuts, and less attention was given to identifying the genes associated with high OA formation. Here, transcriptome analysis was performed to analyze the differentially expressed genes at six different time-points during seed development in normal- and high-oleate cultivars L70 and H176 in order to understand differences in the lipid network between normal- and high-oleic peanut seeds. Our findings provide critical knowledge on the mechanisms involved in OA biosynthesis and regulation in high-oleic peanuts. 

## 2. Results

### 2.1. Changes of Oil Composition during the Developmental Stages of Peanut Seeds

We firstly documented the morphology of developing seeds at six different time points after the gynophore was elongated in the soil in high- and normal-oleic cultivars (Figure 1C). At the early stage (20–30 days after flowering (DAF)), the seed sizes of both peanut cultivars were maintained a lower growth rate and gynophore rapidly swelled to form pods. During this period, the contents of total oil, OA, and LOA in cotyledons did not display an obvious difference between the two cultivars (Figure 1D–F). At the second critical development stage (40–60 DAF), the sizes of seeds and pods continuously increased, and the thickness of pod inner white spongy tissues showed a slight decrease. The contents of total oil in both cultivars showed no difference (Figure 1D), but a significant difference was observed in the contents of OA and LOA between high- and normal-oleic peanut cultivars. The relative proportion of OA increased rapidly (Figure 1E), thus leading to the decreased percentage of LOA in high-oleic cultivar (Figure 1F). In contrast, the relative contents of OA and LOA in the normal-oleic cultivar exhibited no difference at the time points. At the end of seed filling, the seeds gradually stopped growing until maturity, the original spongy tissues disappeared, and the peanut shell was dehydrated to become hard outer shell. In short, OA content in high-oleic cultivar reached the maximum level at the terminal stage of seed growth (Figure 1E), and the relative contents of OA and LOA in normal-oleate cultivar were stable (Figure 1E,F).

The cultivated peanut is a tetraploid species whose genome consists of two diploid subgenomes *Arachis duranensis* (AA) and *Arachis ipaensis* (BB). Therefore, *AhFAD2* contains two homologous genes, *AhFAD2-A* and *AhFAD2-B*, with similar functions. In order to determine whether the *AhFAD2* mutant can cause OA accumulation in the H176 cultivar, we subsequently cloned the full-length sequence (CDS, coding sequence) encoding *AhFAD2* in two peanut cultivars (Appendix A). Sequence alignment indicated four and 12 mutational sites located at *AhFAD2-A* and *AhFAD2-B* encoding region in the H176 cultivar, respectively. The mutation at the 375 bp position caused the original TGT (cysteine) conversion to TGA (stop codon), inducing an early translation termination in *AhFAD2-A* (Figure 1G). Moreover, a single base A was inserted at the position of 442 bp to produce a stop codon (TAA) at the position of 495 bp in the *AhFAD2-B* encoding sequence (Figure 1H), resulting in the termination of translation of *AhFAD2-B*. Therefore, the high concentration of OA in H176 cultivar was attributed to the genotype of *fad2-a*/*fad2-b*.

### 2.2. Transcriptome Assembly

In order to understand the underlying molecular differences between normal- and high-oleic peanuts, we selected six developmental stages of whole seeds for transcriptome sequencing. All of the transcriptome sequence was assembled by a genome-guided strategy, Trinity [22], and individual RNA-seq library data were aligned to *Arachis. duranensis* and *Arachis. ipaensis* genomes (www.Peanutbase.org). The average number of raw reads generated from three biological replicates of each sample was more than 20 million, the average numbers of mapped reads and uniquely mapped reads were far greater than 10 million, and the most mapped ratio of each library to *A. duranensis* and *A. ipaensis* genomes was approximately 50% (Table 1 and Appendix A). After removed the short and redundant reads (Appendix A), the number of obtained transcript assembly contigs (TAC) above 100 bp approximated 257,335, the average length of assembled contig was 1114 bp, and the contig N50 was 1778 bp. Further, the calculated GC content of each library suggested that the average GC content was in the range of 40–50% (Appendix A). Therefore, the assembled transcriptome was able to support our subsequent analysis.

### 2.3. Analysis of Biological Processes in High-Oleic Peanuts

Furthermore, after removing the duplicative gene accession number, we obtained a total of 2138 DEGs in seeds of high-oleic cultivar H176 in comparison with the normal-oleic cultivar L70 at the same developmental stage (Appendix A). The DEG distribution statistics demonstrated that there were 777, 162, 442, 402, 884, and 995 DEGs from stages 1 to stage 6, respectively (Figure 2A). A total of 1,308 up-regulated and 2,354 down-regulated DEGs were identified at six stages in H176 vs. L176 cultivars (Appendix A), indicating that *FAD2* mutation could induce the high-oleic phenotype and suppress generous gene expression. The quantities of identified DEGs in the initial (20–30 DAF) and mature (60–70 DAF) phases were obviously larger than in the middle developmental stage (40–50 DAF), suggesting the involvement of complex transcription regulation events during these stages, particularly the conversion of stearoyl-ACP into OA-ACP in the seed maturity phase. Among the six stages, the middle developmental stage harbored 174 specifically detected DEGs, and 565 DEGs were specifically detected in the stage of early seed swelling, whereas the largest number of specific DEGs (811) was only identified during the period of seed maturity (Figure 2B and Appendix A). In total, 103 DEGs were shared in across three periods (Figure 2B).

The functional interpretation and gene product attributes of these identified DEGs were classified using Gene Ontology (GO) categories. According to the GO analysis, three histograms displayed the top 20 categorized DEG biological processes, molecular functions, and cellular compartments (Appendix A). In terms of biological processes, the metabolic process was the major group, including cellular, primary, protein, and macromolecular metabolism. In terms of molecular functions, at least 200 DEGs were involved in nucleotide and protein binding, and more than 600 DEGs were involved in catalytic activity and binding groups in developing high-oleic peanut seeds. Additionally, most of the DEGs were divided into the top three groups, including the cell, the intracellular area, and the membrane, by cellular compartment analysis (Appendix A). Furthermore, KEGG (kyoto encyclopedia of genes and genomes) pathway enrichment analysis was used to categorize function annotations for all annotated genes. We found that 2138 DEGs were mainly categorized in the top 30 pathways, and the most highly enriched biological pathways in high-oleic peanut seeds were plant hormone signal transduction, starch and sucrose metabolism, phenylpropanoid biosynthesis, and plant–pathogen interaction (Appendix A). In addition, 12 and six DEGs were specifically classified in the pathways for fatty acid metabolism and biosynthesis of unsaturated FAs, respectively (Appendix A). 

### 2.4. Expression Analysis of DEGs

A heatmap representing the differential expressions of a total of 2138 DEGs (Appendix A) in H176 and L70 cultivars was constructed. It revealed the dynamic changes in the transcriptomes during the developmental stages of high-oleic peanut seeds. In the heatmap, the original gene expression data were transformed into log_2_ fold change (Appendix A) and further cluster analysis revealed the significant differences in gene expressions between high- and normal-oleic peanut seeds (Appendix A). In order to characterize these differential expressions in detail, trend analysis was carried out to explore the expression patterns of all DEGs. Total DEGs were clustered into 20 profiles, of which six trend profiles, including 0, 11, 12, 17, 19, and 18 showed significant enrichment (*p* < 0.05) with the colored block, and 14 profiles represented the enrichment of significant trends without color (Appendix A). The expression of 307 genes displayed a reducing trend during the whole seed development in profile 0 and the expression of 129 genes demonstrated an opposite trend in profile 19. The expressions of 140 and 71 genes showed similar trends of increasing at the early stages in profiles 17 and 18, respectively. In profile 11, the expression of 232 genes showed an initial increase but subsequent reduction at stage 2 (30 DAF) to stage 3 (40 DAF) in developing seeds. However, the expressions of 206 genes exhibited only increases during the period of initial and terminal stages in profile 12 (Appendix A). 

### 2.5. Dynamic Changes of DEGs Related to Lipid Metabolism in High-Oleic Peanut

Negative variation in *FAD2* of H176 cultivar promoted OA accumulation, but suppressed the production of PUFAs (polyunsaturated fatty acids). Here, DEGs associated with lipid biosynthesis and metabolic pathways were identified. Specifically, a total of 74 DEGs were characterized as involved in lipid metabolic pathway at all development stages of seeds (Table 2). Further GO analysis suggested that at least 10 genes were classified into a series of primary processes, such as lipid metabolism and biosynthesis, cellular lipid metabolism, and FA biosynthesis and degradation. KEGG enrichment indicated that the *FAD2* mutation-associated high-oleic content contributed greatly to the gene expression patterns of FA and glycerolipid synthetic or metabolic pathways (Figure 3A and Appendix A). From the holistic perspective, the heatmap displayed differences in those gene expression patterns between high- and normal-oleic groups according to the shaded color blocks (Figure 3B). Moreover, trend analysis was performed to reveal the expression patterns of all lipid DEGs. Total DEGs were clustered into top 10 profiles, of which three trend profiles, including profiles 0, 7, and 9, were significantly enriched (*p* < 0.05) (colored block), and seven profiles represented the enrichment of significant trends (without color) (Figure 3C). The expression of 14 genes displayed a declining trend during the whole seed development stage in profile 0, but the expressions of nine genes demonstrated an opposite trend in profile 9. In profile 7, the expression of nine genes exhibited an obvious increase during the initial and terminal stages (Appendix A).

Moreover, to explore the existing annotation information of all lipid DEGs more deeply, the InterPro web-database (www.ebi.ac.uk/interpro) was used to analyze and explain the molecular functions of lipid genes. Based on the valid functional unscramble of InterPro, a majority of DEG-encoded proteins contained the conversed domain of acyltransferase, whose molecular functions were tightly linked to FA biosynthesis–metabolism, FA oxidation–reduction, FA elongation, lipid oxdiation, and the glycerolipid biosynthesis pathway. Interestingly, five DEGs were found to encode plant *fatty acid desaturase*, including *Aradu.AV1HQ*, *Aradu.XM2MR*, *Araip.S3GXY*, *Aradu.PR2BP*, and *Araip.31Q5V* (Table 2). Among these identified *fatty acid desaturease* genes, *Aradu.XM2MR* (*SAD2*) encodes a *stearoyl-acyl carrier protein desaturase* whose relative expression specifically increased at the early stage (20–30 DAF) but decreased at the terminal stage (60–70 DAF) during the development of high-oleic peanut (H176) seeds. On the other hand, the expression of other four *fatty acid desaturase* genes were abnormally down-regulated at distinct time points, and their expression levels were inhibited due to the *fad2* mutation-induced OA accumulation. Therefore, we concluded that *fad2* repressed the expressions of multiple members of *fatty acid desaturase* family based on transcriptome analysis.

### 2.6. Analysis of the Expression of FADs at the mRNA level

Additionally, to validate whether the difference in RNA-seq levels of lipid DEGs is truly reflecting the actual transcription level, quantitative real-time PCR (polymerase chain reaction) was conducted for detecting the expression levels of DEGs at six consecutive periods between H176 and L70 cultivars. Indeed, the relative expression levels of *fatty acid desaturase* family genes (*Aradu.AV1HQ*, *Aradu.XM2MR*, *Araip.S3GXY*, and *Araip.31Q5V*) were consistent with the dynamic changing trend of RNA-seq obtained RPKM (reads per kilobase per million) values (Figure 4 and Appendix A), but the opposite results were obtained from the examination of *Aradu.PR2BP* along with seed development. In particular, the expression of *Aradu.XM2MR* presented a sudden increase at the early stage (20–30 DAF) and then a gradual drop until the seed maturity in the H176 cultivar. Therefore, we concluded that *FAD2* mutant-induced high oleate content affected the expression patterns of *SAD2* (*Aradu.XM2MR*) at the mRNA level.

### 2.7. Subcellular Localization of FAD2 and SAD2 (Aradu.XM2MR) 

*FAD2* mutation induced high-oleic feedback mechanisms to repress the transcription of upstream *SAD2* (*Aradu.XM2MR*); however, whether *SAD2* (*Aradu.XM2MR*) is also localized to the endoplasmic reticulum (ER) as *FAD2* remains unclear. We showed that the green signal from *SAD2*-GFP (green fluorescent protein) when overlaid with red chloroplast marker presented an obvious yellow signal in *Arabidopsis* protoplast cells, suggesting that *SAD2* was localized to chloroplast in the peanut (Figure 5A,B). Additionally, the conversed *fatty acid desaturase* domain of *FAD2* (Appendix A) indicated that it was likely localized to the endoplasmic reticulum like its homologous gene in *Arabidopsis*, but it remains unclear whether the *FAD2* targets the ER in peanut. We transiently co-expressed the *FAD2*–GFP fusion protein and ER marker in *Arabidopsis*’ protoplast cells. As expected, the green signal of the anchored organelle of *FAD2*-GFP was completely co-localized with red ER marker in *Arabidopsis* protoplast cell (Figure 5C), which suggested that *FAD2* was localized to the ER in peanut. Subcellular localization analysis demonstrated that critical enzymes of oleic acid biosynthesis were located at different organelles; upstream *SAD2* regulated the conversion of stearoyl-ACP (C18:0-ACP) to oleoyl-ACP (C18:1-ACP) in the chloroplast, but the downstream *FAD2* mutation caused defects of oleic acid (C18:1) conversion to linoleic acid (C18:2) in the ER. 

Further phylogenetic analysis by subjecting the homologous sequence of *stearoyl-ACP desaturase 2* in peanut genome to BLAST (basic local alignment search tool) analysis suggested that *SAD2* (*Aradu.XM2MR*) exhibited high similarity in evolution with *ahay.7G97IV* and *ahay.H654N0* (Figure 6A), as well the conserved *fatty acid desaturase* domain (56–413 amino acids) in the *Aradu.XM2MR* protein sequence (Figure 6B). Additionally, ectopic expression of *Aradu.XM2MR* in yeast cells increased the relative intensity of C18:1 in the LC-MS (liquid chromatograph mass spectrometer) profile (Figure 6C and Appendix A), suggesting that the *Aradu.XM2MR*-encoded product could transform C18:0 to C18:1 like the original *SAD2*. 

### 2.8. The Enhanced Activity of Peroxidase (POD) in High-Oleic H176

Peroxidase (POD) plays an important role in the metabolism of fatty acids, which takes place in the peroxisome. Therefore, we extracted and examined the POD activity at six different time points during seed development. In high-oleic cultivar H176, the POD activity was up-regulated by two-fold at 30–40 DAF (Figure 7A). However, with the seed maturation, POD activity was hardly detected in normal- and high-oleic cultivars. This finding suggested that high-oleic H176 induced up-regulation of POD activity before seed maturation in comparison with normal-oleic cultivar L70. Furthermore, the H_2_O_2_ content in both peanut cultivars were examined, which presented a positive correlation with POD activity at 30–40 DAF during seed filling, and the concentration of H_2_O_2_ in the high-oleic cultivar was higher than that in the normal cultivar L70 (Figure 7B). Twelve DEGs of the POD family were identified in high-oleic H176, implying that the excessive accumulation of high-oleic acid triggered the POD-mediated β-oxidation (Figure 7C). To understand whether the expression level of *SAD2* was affected by ROS (reactive oxygen species), we utilized the exogenous H_2_O_2_ to treat the normal-oleic cultivar (L70) seedlings. This result indicated that H_2_O_2_ was capable of triggering the expression of *SAD2*, but high content of H_2_O_2_ displayed an inhibitory effect (Figure 8A). According to our study, the redox signaling caused by POD likely acted as a secondary messenger to regulate fatty acid metabolism.

## 3. Discussion

Transcriptome sequencing provides a systemic approach for studying gene expression and network interactions in different organs of developing peanut seeds. Herein, by generating a high-quality transcriptome dataset of high-oleic and low-oleic peanut seeds, we obtained 74 specific DEGs (Table 2) involved in the biosynthesis or metabolic pathways of fatty acid, lipid, phospholipid, sphingolipid, and glycerolipid. Based on the analysis of the gathered transcriptome data of the FA synthesis pathway, we concluded that *FAD2* mutation-caused high-oleic feedback regulates the expression patterns of upstream *oleoyl-ACP (C18:1-ACP) synthetase* gene *SAD2* (*Aradu.XM2MR*) [23] at the mRNA level during the development stages of high-oleate peanut seeds. Meanwhile, *fad2-a/fad2-b* variation with a high content of OA not only affected upstream *SAD2* (*Aradu.XM2MR*) expression, but also directly changed the level of downstream *linolenic acid (C18:3) synthase FAD8* [24] and *linoleic acid (C18:2) synthetase delta−6 desaturase* (*FAD6*) [25] (Appendix A). Chloroplastic enzyme *FAD6* is responsible for the biosynthesis of 16:2 and 18:2 FAs from galactolipids, sulpholipids, and phosphatidylglycerol using ferredoxin as the electron donor, and its mutation in plants could result in a reduced level of unsaturated fatty acids, thus leading to susceptibility to photo-inhibition [26]. *FAD8* encodes *omega−3 fatty acid desaturase*, and gene mutation could lead to the reduction in desaturation of storage and membrane lipids [27]. In conclusion, *FAD2* mutation prevented the carbon flux of unsaturated fatty acid synthesis, thus correspondingly resulting in the down-regulated transcription of *FAD6* and *FAD8* (Appendix A). These results allowed us to propose a strategy for improving the content of unsaturated FA in the peanut by manipulating the FA desaturase family of genes. 

Our results are consistent with a previous comparative proteomics study to dissect the high-oleate-suppressed peanut *SAD2* activity [28]. Our finding provides new insight into understanding the mechanism of OA synthesis in the peanut, and showed that over-accumulated OA, as the downstream reaction product, repressed the transcription of upstream rate-limiting enzyme *SAD2* (*Aradu.XM2MR*) along with the maturation of the seeds. Additionally, the majority of *SAD* enzymes [29,30] are usually predicted to be located at the plastid (dark condition) and chloroplast (light condition), and are responsible for catalyzing the substrate stearoyl-ACP (C18:0-ACP) to form oleoyl-ACP (C18:1-ACP) in the *de novo* plant FA synthesis. Subsequently, *acyl-ACP thioesterase* (*FATA*) [31] suppresses the ACP domain and accelerates the conversion of acyl-ACP fatty acid to free fatty acid. Finally, the released acyl-oleic acid (Acyl-C18:1) is exported from the cytoplasmic plastid into the endoplasmic reticulum (ER) by the acetyl coenzyme A transport pathway in order to synthesize linoleic acid or promote the flux of the 18-carbon atom chain (C18) for conversion into other unsaturated FAs [32,33]. *FAD2* is mainly anchored in ER and catalyzes the transformation of OA into LOA, whereas the allelic mutant displays defects in this conversion, thus inducing high-oleic content [34]. It is well known that the regulation of transcription is dependent on intra-nuclear transcription factors [35]. A hypothesis of the decreased *SAD2* (*Aradu.XM2MR*) mRNA level in the high-oleic peanut cultivar is attributed to uncharted transcriptional repressors. The excessive accumulation of OA could trigger multiple signal pathways closely connected with a relationship triangle of chloroplast–ER–nucleus-mediated signaling transduction for coordinated inhibition of *SAD2* (*Aradu.XM2MR*) expression (Figure 8B). 

Alternatively, mature plant seeds store a diversity of FAs that are re-disintegrated to supply nutrients for the growth of post-germinative seedlings [36]. Once the relative content of OA exceeds a normal threshold in total reserved compounds, the feedback signaling can be detected by the peanut, and the bio-synthetic network of OA will down-regulate upstream reaction *via* repressing *SAD2* (*Aradu.XM2MR*) expression. Meanwhile, we showed evidence that over-accumulation of OA can trigger the degradation of fatty acid by β-oxidation in the peroxisome thus generating ROS-modulated signaling transduction between *FAD2* and *SAD2* (*Aradu.XM2MR*). Free fatty acids as substrates are imported into peroxisome by an ABC (ATP-binding cassette) transporter *CTS* (*COMATOSE*) [37,38]. In addition, 12 DEGs relevant to the function of POD were identified in high-oleic cultivar H176, and POD activity and H_2_O_2_ content showed significant increases in the early period of seed development, which exhibited the same trend as the expression pattern of *SAD2* (*Aradu.XM2MR*) in high-oleic H176. Accordingly, *FAD2* mutation mediated over-accumulation of oleic acid likely triggered a redox signaling burst that correlated with the peroxidase-mediated FA degradation (Figure 7C). Additionally, a series of DEG-encoded enzymes are involved in FA oxidation-reduction process, such as *trans-2-enoyl-CoA reductase*, *acetyl-CoA carboxylase*, *sterol C4-methyl oxidase*, *lipoxygenase*, and *seed linoleate 9S lipoxygenase* (Table 2) [39,40,41,42]. 

Theoretically, the dynamic change of *FAD2* should be easily detected in the transcription profile due to the novel allelic mutation, and the predictable change was found at each period of seed developmental in the high-oleic peanut cultivar H176. However, our previous results indicated that the *FAD2* protein was maintained at a relatively stable level for OA accumulation cross the entire time during the developmental stages of high-oleic peanut seeds [28]. Herein, an interpretation of the relevant phenomena from transcriptome-seq was based on the finding that *FAD2* mRNA (*Araip.S3GXY*) was not able to normally transcribe along with the high-oleic seed maturation, thereby leading to the significantly declined transcription level when compared with normal cultivar, and the *FAD2* enzyme lost its activity due to premature termination (Figure 1G,H). Furthermore, high-oleic acid still can affect the expression levels of some chloroplast enzymes involved in FA biosynthesis process. For instance, *3-ketoacyl-CoA synthase* (*KCS*) 11 can catalyze the first step during the fatty acid elongation process [43], which is the condensation of C2 units to acyl-CoA, and the peak value of *KCS11* expression presented at stage 3. The *3-oxoacyl ACP synthase 3* (*FabH*) regulates malonyl-ACP in the presence of acyl-CoA to generate 3-ketoacyl-CoA [44]. Both enzymes exhibited consistent expression patterns. *FabZ*-encoded *dehydrogenase* promotes the conversion of hydroxyacyl-ACP to enoyl-ACP [45], whose expression showed a two-fold increase at stage 3 and at the terminal phase (Appendix A). Taken together, a comprehensive overview of *FAD2* mutant-induced expression network is provided, and the transcriptome-identified DEGs offer a potential resource to uncover the mechanisms of the FA synthesis pathway in different organelles. Moreover, the possible functions of those lipid DEGs will be further elucidated in coming investigations for dissecting OA-associated FA biosynthesis in the peanut.

## 4. Materials and Methods

### 4.1. Plant Materials

Two peanut cultivars H176 (Kainong176) and L70 (Kainong70) (Figure 1A,B) with different OA contents in their mature seeds (Kainong176 > 70% of total oil, Kainong70 < 40% of total oil) were used in the present study [28]. High-oleic-content mutant lines (Kainong176 and Kainong70) were isolated from the population of sexual hybridization by using Kainong30 (♀) and Kaixuan01−6 (♂) at Kaifeng (Henan Province, China) (Appendix A). These two cultivars were provided by Kaifeng Academy of Agriculture and Forestry, and they were re-cultivated to flowering in the field of Baiyun District of Guangzhou City (Guangdong Province, China) under normal growth conditions. Subsequently, the aerial pegs were labeled with a pink tag after flowering, and secondary labeling was conducted at the time when the elongated gynophore first touched the ground [28]. The developmental stages of those marked gynophores were precisely recorded as days after flowering (DAF) in high- and normal-oleic cultivars. Non-elongated first-marked gynophores were discarded. When the gynophores entered into the soil and developed into pods, we collected the seeds at different times (from 20 to 70 DAF). In order to improve consistency, we chose pods with similar phenotypes at the same developmental stage for RNA analysis. The harvested seeds were quickly frozen in liquid nitrogen, and their total FA and RNA were extracted for relevant experiments. 

### 4.2. Analysis of OA, LOA, and Oil Content

Oil content and composition were analyzed using 5 g of seeds collected from each developmental stage. The fresh seeds of different stages were ground into a fine powder in liquid nitrogen. The total oil content was determined following Soxhlet extraction using n-hexane. The molecular composition of extracted FAs was determined by gas chromatography of the methyl ester of the FAs according to the National Standard of the People’s Republic of China (GB/T 17377−2008). Briefly, 100 mg peanut seeds were frozen and ground into a fine powder in liquid N2. The powder was mixed with 1.5 mL chloroform/methanol (2:1, *v*/*v*) and 100 μL internal standard solution C17:0 (1 mg/mL). The mixture was extracted by ultrasound-assisted extraction for 30 min and centrifuged at 12,000 rpm for 6 min, and then 1 mL supernatant was transferred into a fresh tube. The FA extracting solution was mixed with 0.2 mL KCl (75%, *w*/*v*) solution and centrifuged at 12,000 rpm for 6 min, and then 400 μL substratum chloroform extract phase was transferred into a new glass tube and prepared for methanol esterification. The extracting solution was mixed with 2 mL sulfuric acid in methanol (5:95, *v*/*v*) and incubated in a water bath at 85 °C for 1.5 h. Subsequently, 1 mL H_2_O and 1 mL hexane were added into the tube, the mixture was centrifuged at 5000 rpm for 5 min, and 500 μL superstratum hexane extraction phase was used for gas chromatography (Cat#YLSB076). The individual FA contents were reported as relative percentages of OA and LOA in the extracted oil [46].

### 4.3. Total RNA Isolation, Library Construction, and Transcriptome Sequencing

Total RNA was extracted from 300 mg of seeds from both peanut cultivars using the Trizol Reagent (Invitrogen, Beijing, China). The concentration of purified RNA was determined using a NanoDrop UV-visible spectrophotometer (Thermo Fisher, Waltham, MA, USA). Total RNA was used for library construction following the High-Throughput Illumina Strand-Specific RNA Sequencing Library protocol. Each library was made with three individual peanut seed samples, and each library was subjected to analysis with six replicates. The mRNA for sequencing was purified from 5 µg total RNA using Agencourt Ampure-XP magnetic beads (Beckman Coulter, Brea, CA, USA), and the purified mRNA was fragmented to the average size of 300 bp. The first-strand cDNA was synthesized using the fragmented mRNA as the template with random hexamer primers. Second-strand cDNA synthesis was performed by using DNA polymerase I and RNase H. All RNA-Seq libraries were sequenced on a Illumina HiSeq™2000, and all the transcriptome raw data have been deposited in Sequence Read Archive (SRA) database with the BioProject Accession number: PRJNA480751 [47].

### 4.4. Bioinformatic Analysis of Transcriptome Data

RNA-library sequencing and transcript assembly were performed by Novogene Ltd. based on the Trinity [22] method. The Trinity method consisted of three software modules, *Inchworm*, *Chrysalis*, and *Butterfly*, that were applied sequentially to process large volumes of RNA-Seq reads. Assembled reads were mapped to the published genome of two diploid ancestors *Arachis duranensis* and *Arachis ipaensis* (www.Peanutbase.org). Reads per kilobase per million (RPKM) were calculated by adopting an in-house script according to the count table of assembly sequences output. The differentially expressed genes (DEGs) were identified by applying a cutoff log_2_ (fold change) of >1 and <−1 for up-regulated and down-regulated genes, respectively, as well as *p* < 0.05. Gene Ontology analysis of DEGs was performed by AgriGO web-based tools. KEGG pathway enrichment was assigned to the DEGs using the online KEGG web server. 

### 4.5. Real-Time PCR Analysis

Real-time PCR was carried out according to a previously described method [48]. Briefly, 1 µg of total RNA was reversely transcribed into cDNA using the PrimeScript RT Kit (Takara, Dalian, China) according to the manufacturer’s instructions. The PCR reaction was conducted on a 20-µL scale using SYBR Premix ExTaq™ (TaKaRa, Dalian, China) on an ABI (Applied Biosystems) StepOne Plus system. The relative expression of target genes was calculated by the 2^−ΔΔ*C*t^ method and shown as fold changes relative to the normal-oleate cultivar kainong70. *Yellow leaf specific 8* gene (*yls8*) (Forward Primer: 5′-AACTGCTTAGCTGCTATTACC−3′, Reverse Primer: 5′-TCGCCAAATAACACGTTGCAT−3′) and *alcohol dehydrogenase class III* (*adh3*) (Forward Primer: 5′-GACGCTTGGCGAGATCAACA−3′, Reverse Primer: 5′-AACCGGACAACCACCACATG−3′) were selected as the internal controls (Appendix A). Each measurement was carried out in triplicate with three replicates, and data were expressed as means ± SE.

### 4.6. Subcellular Localization Analysis

The normal full-length *FAD2* [10] and *Aradu.XM2MR* cDNA without a stop codon were cloned between the cauliflower mosaic virus (CaMV) 35S promoter and the GFP gene. The resulting *FAD2*/*Aradu.XM2MR*-GFP fusion construct and different organelle markers were transiently co-expressed in *Arabidopsis* protoplasts by PEG induction [49]. Briefly, *Arabidopsis* protoplast cells were isolated from two-week-old *Arabidopsis* leaves. The full leaves were cut into approximately 0.5−1 mm strips, and incubated in 10 mL enzyme solution (1.5% Cellulase RS, 0.75% Macerozyme R−10, 0.6 M mannitol, 10 mM MES at pH 5.7, 10 mM CaCl_2_, and 0.1% BSA) for 4 h in the dark with gentle shaking (60 rpm). After washing twice with W5 solution (154 mM NaCl, 125 mM CaCl_2_, 5 mM KCl and 2 mM MES at pH 5.7) and centrifuging at 100 g at 4 °C for 2 min, the supernatant was discarded. Protoplast cells were resuspended in MMG solution (0.4 M mannitol, 15 mM MgCl_2_ and 4 mM MES at pH 5.7). Then, 100 μL of the protoplast cell solution with 10 μg plasmid and 110 μL 40% PEG 4000 were incubated at 28 °C for 15 min in the dark, followed by the addition of 1 mL W5 and centrifugation at 150 *g* at 4 °C for 2 min. The supernatant was discarded. The collected protoplast cells were resuspended in 500 μL WI solution (0.5 mM mannitol, 4 mM MES at pH 5.7 and 20 mM KCl) and incubated at 28 °C for 12−18 h in dark. Next, the protoplast cells were collected by centrifugation at 300 *g* at 4 °C for 10 min and resuspended in 40 μL WI solution for confocal microscopy. Fluorescence was examined under a laser-scanning confocal microscope (Model LSM 780; Carl Zeiss, Jena, Germany).

### 4.7. POD Activity Assay

Fresh peanut seeds (300 mg) were frozen and ground in liquid nitrogen. The powder was mixed with 4 mL 0.05 mol/L PBS (pH 7.8) and transferred into a 5-mL tube. After thawing, the tubes were centrifuged at 8000 rpm/min for 15 min, and the supernatant containing the total peroxidase was collected. The POD activity was measured as the rate of decomposition of H_2_O_2_ with guaiacol as the hydrogen donor by spectrophotometrically measuring the rate of color development at 436 nm due to guaiacol oxidation [50]. H_2_O_2_ content was determined by following the method described by Sui et al. (2018) [47]. The concentration of H_2_O_2_ was estimated by detecting the absorbance of the titanium-hydroperoxide complex at 415 nm after calculation with a standard curve plotted with H_2_O_2_ standards [47].

### 4.8. Ectopic Expression of SAD2 (Aradu.XM2MR) in Yeast Cells and Fatty Acid Composition Analysis

The protocol of ectopic expression in yeast cells was reported by Chi et al. (2011) [23]. Briefly, the *SAD2* CDS sequence was cloned into the vector pYES2, and the CK group (empty vector pYES2) and pYES2-*SAD2* were co-transformed into the yeast strain INVSc1 by utilizing the lithium acetate method. Positive yeast clones were selected on a medium plate (SD/-Ura), and the positive yeast clones were cultivated at 30 °C for 36 h to examine the fatty acid composition. LC-MS analysis of the fatty acid composition was performed by ultra-performance liquid chromatography coupled with an electrospray ionization–quadrupole time-of-flight mass spectrometer system. This experiment and relevant data analysis were performed by Deepxomics Co., Ltd., Shenzhen, China. 

## Figures and Tables

**Figure 1 ijms-20-03091-f001:**
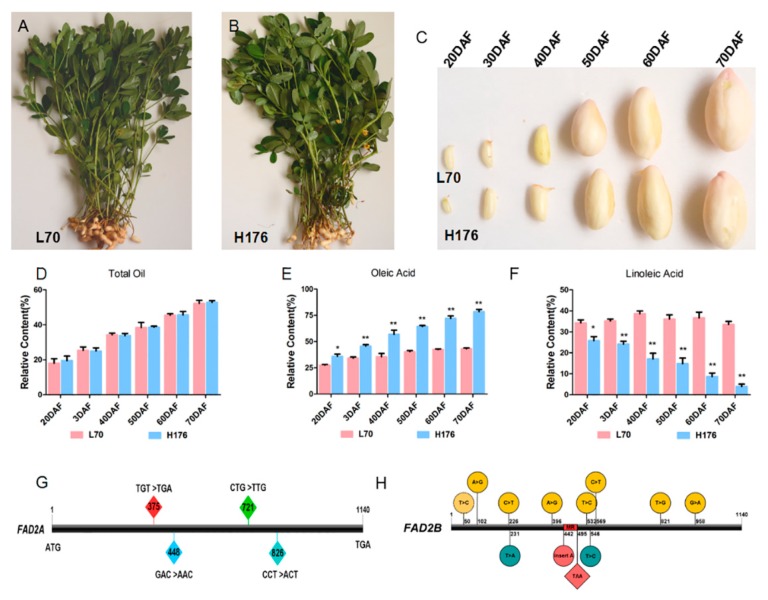
Morphology and fatty acid profiles during the high- and normal-oleic peanut seed development. (**A**,**B**) Phenotypes of L70 and H176 plants. (**C**) Seed samples harvested from six different development stages (20–70 days after flowering (DAF)) in L70 and H176. (**D**–**F**) These histograms display the dynamic changes of total oil, oleic acid, and linoleic acid along with seed development. Values are shown as the mean (±SD) of three biological replicates, and asterisks indicate a significant difference (* *p* < 0.05, ** *p* < 0.01) when compared with the normal-oleic peanut variety L70. (**G**) Four mutant sites located at *FAD2-A* coding region in H176 to generate a stop codon (TGA) at the 375 bp position, inducing the termination of protein synthesis. (**H**) Gene structure of *FAD2-B* in H176. Twelve mutant sites exist in the coding sequence. An adenine was inserted at the 442 bp position, thus generating a stop codon (TAA) at the 495 bp position. MR: mutant region.

**Figure 2 ijms-20-03091-f002:**
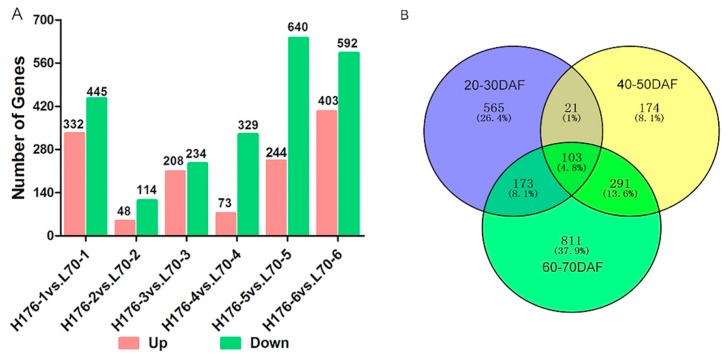
Statistical analysis of all identified differentially expressed genes (DEGs) in L70 and H176. (**A**) Statistical analysis of differentially expressed genes (DEGs) at different seed development stages in H176 vs. L70. (**B**) Venn diagram showing the distribution of DEGs at three periods of seed development in H176 vs. L70. The blue circle indicates the 20–30 DAF, the yellow circle indicates the 40–50 DAF, and the green circle indicates 60–70 DAF.

**Figure 3 ijms-20-03091-f003:**
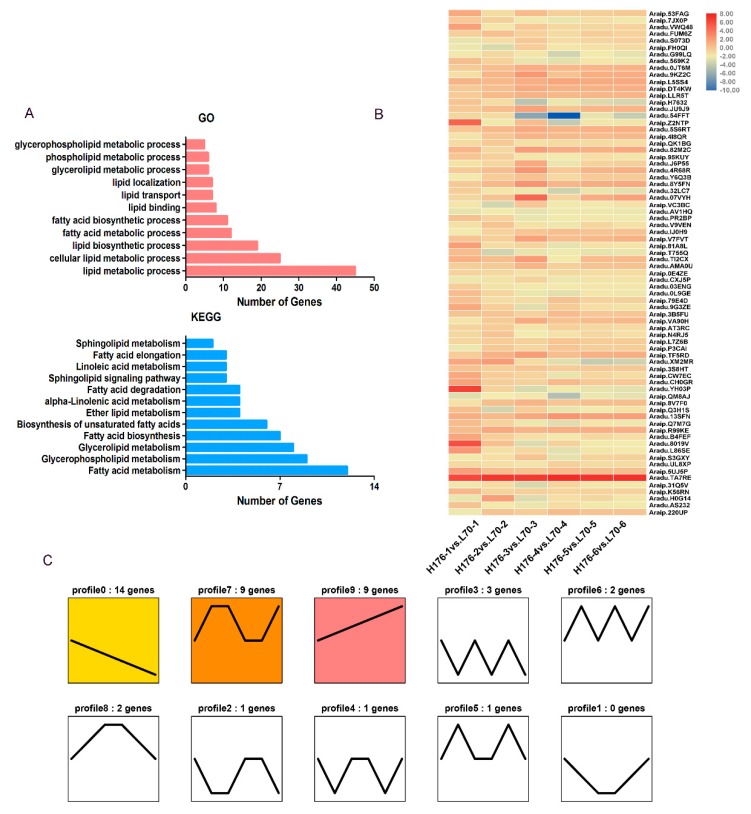
Expression analysis of lipid DEGs among six developmental stages in h176 vs. L70. (**A**) GO (gene ontology) and KEGG (kyoto encyclopedia of genes and genomes) enrichment analysis of total 74 lipid DEGs. (**B**) Heatmap showing the relative expression of total lipid DEGs at each stage in L70 and H176. (**C**) Trend analysis of lipid DEG expression (10 trends). Colored block trend: significant enrichment trend (*p* < 0.05). Without color trend: the enrichment of significant trends.

**Figure 4 ijms-20-03091-f004:**
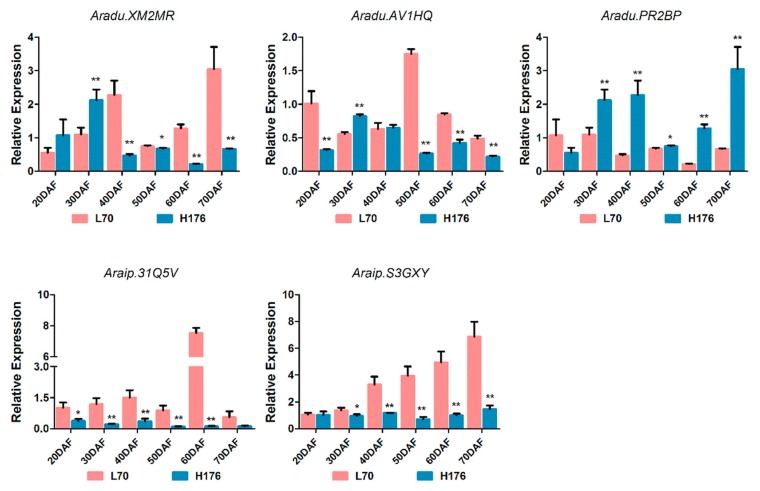
Quantitative real-time PCR (polymerase chain reaction) validated the relative expression levels of lipid DEGs at the transcriptional level in L70 and H176. Each measurement with three biological replicates, and the values are shown as means±SD (* *p* < 0.05, ** *p* < 0.01) compared with normal-oleate cultivar L70.

**Figure 5 ijms-20-03091-f005:**
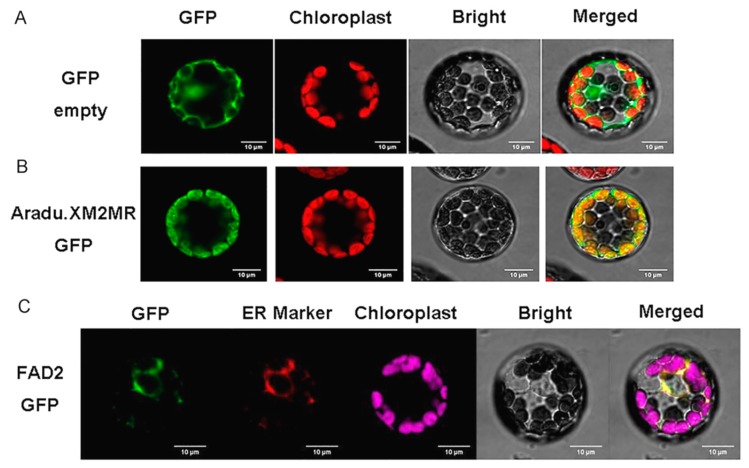
Subcellular localization analysis of *FAD2* and *Aradu.XM2MR* in the *Arabidopsis* protoplast cell. (**A**) An empty GFP vector served as the control. (**B**) *Aradu.XM2MR*-GFP (green) merged with chloroplast marker (red) to generate a yellow signal. (**C**) *FAD2*-GFP (green), endoplasmic reticulum (ER) marker (red), chlorophyll autofluorescence (pink), and the bright field image were recorded and the resulting images were merged. Scale bar is 10 µm.

**Figure 6 ijms-20-03091-f006:**
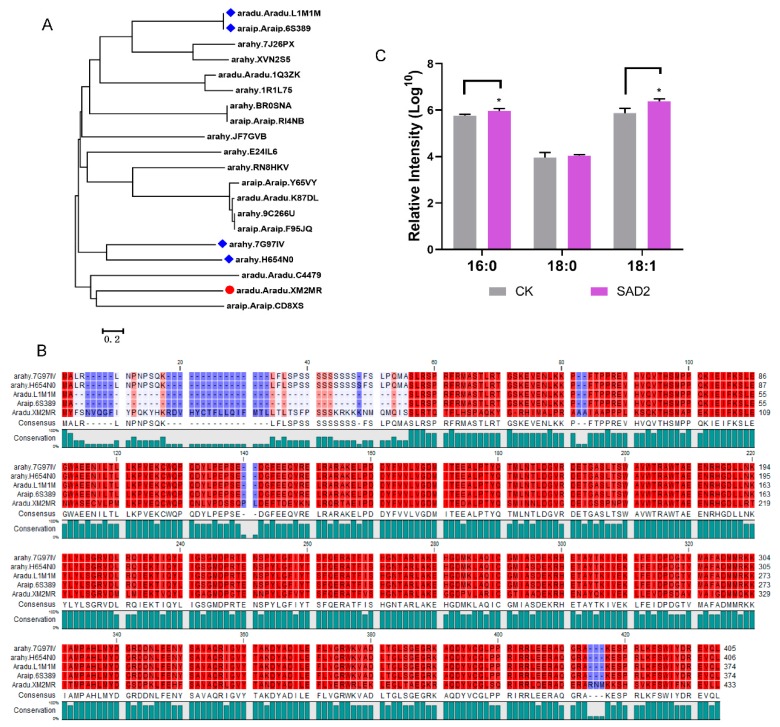
Sequence analysis of *SAD2* (*Aradu.XM2MR*). (**A**) Comparative phylogenetic analysis of the *SAD2* homologous proteins in peanut. The phylogenetic tree was generated by a Neighbor-Joining algorithm with the software MEGA6 (bootstrap test, 2000 replicates). Gene accession number referenced the peanut database (www. PeanutBase.org). (**B**) Protein sequence alignment of *SAD2* (*Aradu.XM2MR*) by the software CLC sequence viewer 7. (**C**) Histogram displaying the relative intensity (log_10_) of C16:0, C18:0, and C18:1 in yeast cells expressing *SAD2* (*Aradu.XM2MR*), respectively. CK indicates empty vector pYES2. *SAD2* indicates the pYES2-*Aradu.XM2MR*. Values shown are means±SD of three biological replicates, and asterisks indicate a significant difference as compared with the control group (* *p* < 0.05).

**Figure 7 ijms-20-03091-f007:**
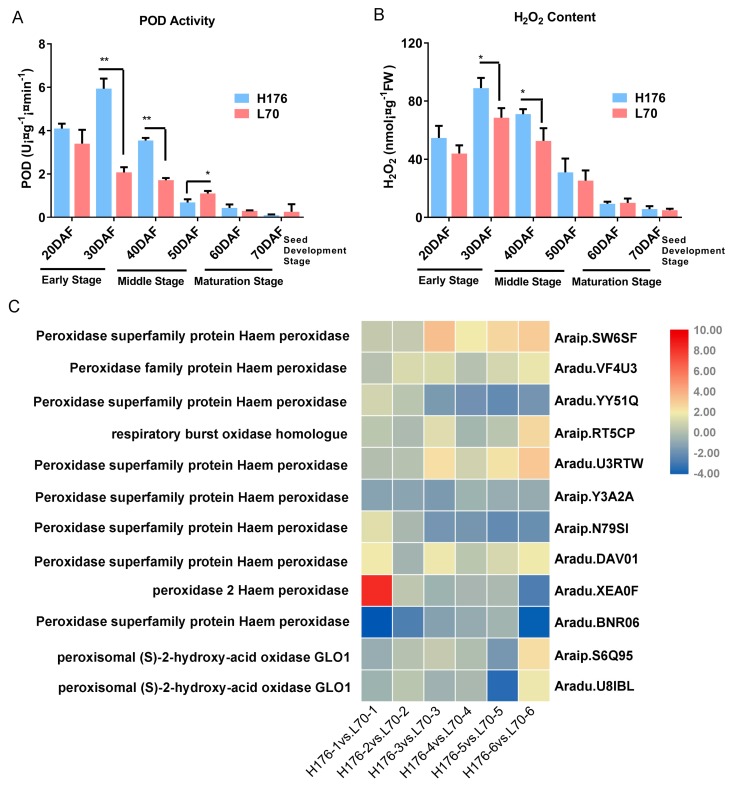
The POD activity and H_2_O_2_ content in the peanut seed. (**A**) POD activity in normal- and high-oleic peanut seeds. (**B**) H_2_O_2_ content in normal- and high-oleic peanut seeds. Values shown are means±SD of three biological replicates, and asterisks indicate a significant difference compared with normal-oleic control L70 (* *p* < 0.05, ** *p* < 0.01). (**C**) Heatmap displaying the expression difference of POD DEGs in the developing seeds of the high-oleic seeds. Each block represents the log_2_ fold change value of H176 vs. L70.

**Figure 8 ijms-20-03091-f008:**
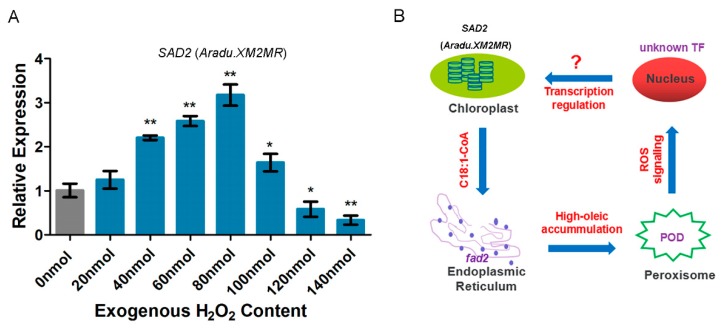
Putative model of *FAD2* mutation-induced feedback in regulating the expression of upstream oleic-synthesis gene *SAD2* (*Aradu.XM2MR*). (**A**) Examination of *SAD2* (*Aradu.XM2MR*) expression level by using the exogenous H_2_O_2_ treated the normal-oleic cultivar seedlings. Values shown are means±SD of three biological replicates, and asterisks indicate a significant difference (* *p* < 0.05, ** *p* < 0.01) compared with the control group (0 nmol). (**B**) *FAD2* mutation located at the ER membrane. High-oleic accumulation in the peroxisome triggers FA β-oxidation reaction, and then ROS signaling mediates nucleus’s transcription factors (TFs) to regulate the transcription level of chloroplast (or plastid, under dark condition) protein *SAD2* (*Aradu.XM2MR*) via an unknown signaling pathway.

**Table 1 ijms-20-03091-t001:** Statistical analysis of RNA-seq raw-reads mapped to the ancestor genome.

Samples	Average Raw Reads	Average Mapped Reads	Average Multiple Mapped Reads	Average Uniquely Mapped Reads	Average Mapped Ratio (%)
Reads mapped to *Arachis duranensis* (AA) genome					
H176	22,918,504.50	13,166,271.00	360,387.50	12,805,882.83	57.33
L70	25,944,882.83	14,650,167.00	331,180.67	14,318,985.50	56.33
Reads mapped to *Arachis ipaensis* (BB) genome					
H176	22,918,504.50	13,441,533.00	272,607.00	13,168,925.83	58.67
L70	26,278,776.33	15,236,358.17	310,790.83	14,925,567.50	57.83

**Table 2 ijms-20-03091-t002:** Identification of lipid DEGs at each seed development stages in H176 vs. L70.

Gene Name in Diploid Ancestor	Gene Name in Cultivated Peanut	Log_2_ Fold Change						Function Description
		Stage 1 H176-1 vs. L70-1	Stage 2 H176-2 vs. L70-2	Stage 3 H176-3 vs. L70-3	Stage 4 H176-4 vs. L70-4	Stage 5 H176-5 vs. L70-5	Stage 6 H176-6 vs. L70-6	
*Araip.53FAG*	*Arahy.6ZS5B3*	1.39	−1.1	0.47	−0.56	−0.43	−0.23	aldehyde dehydrogenase family 3 member F1-like
*Araip.7JX0P*	*Arahy.7L16MH*	−0.2	−0.61	−1.87	−2.35	−1.34	−1.8	triacylglycerol lipase
*Aradu.VWQ48*	*Arahy.94LY4L*	1.85	−1.77	−0.21	−1.04	−0.98	−0.69	lipid transfer protein
*Aradu.FUM6Z*	*Arahy.R5P48D*	−0.56	0.19	−0.55	−1.14	−0.39	−0.7	UDP (uracil 5′-diphosphate) -glycosyltransferase superfamily protein
*Aradu.S073D*	*Arahy.RJ9TTJ*	−2.41	−1.25	−0.26	−1.24	−0.84	−0.73	senescence-associated carboxylesterase 101-like isoform X2
*Araip.FH0QI*	*Arahy.W0JK2Q*	−2.16	−3.06	−0.45	−1.74	−1.45	−1.12	phosphatidylinositol-4-phosphate 5-kinase family protein
*Aradu.G99LQ*	*Arahy.ZS7YSR*	−1.86	−0.99	−1.64	−3.45	−1.98	−2.53	seed linoleate 9S-lipoxygenase
*Aradu.569K2*	*Arahy.LS6X36*	0.31	−0.71	−2.37	−2.68	−1.73	−2.22	phosphoinositide phospholipase C 6-like
*Aradu.0JT6M*	*Arahy.KBA53Y*	−0.33	0.28	1.21	0.63	0.73	0.86	biotin carboxyl carrier acetyl-CoA carboxylase
*Aradu.9KZ2C*	*Arahy.UBEE9G*	−1.37	0.78	2.29	0.15	1.13	1.21	3-oxoacyl-(acyl-carrier-protein) synthase 3 FabH
*Araip.L5SS4*	*Arahy.G623B4*	0.93	0.65	1.24	1.66	1.34	1.42	GDSL(Gly-Asp-Ser-Leu) -like lipase/acylhydrolase superfamily protein
*Araip.DT4KW*	*Arahy.R4EZU5*	−0.27	0.31	1.39	0.96	0.98	1.13	1-acyl-sn-glycerol−3-phosphate acyltransferase-like protein
*Araip.LLR5T*	*Arahy.H4YX61*	−0.36	0.51	0.99	0.66	0.73	0.8	biotin carboxyl carrier acetyl-CoA carboxylase
*Araip.H7632*	*Arahy.Y66AIQ*	0.02	−1.23	−4.48	−2.27	−3.01	−3.3	1-phosphatidylinositol phosphodiesterase-like protein
*Aradu.JU9J9*	*Arahy.4ZRU7U*	−0.3	0.37	1.81	−0.04	0.72	0.83	trans-2-enoyl-CoA reductase, steroid 5-alpha-reductase
*Aradu.54FFT*	*Arahy.DCBE83*	−0.99	−1.1	−6.54	−11.5	−2.45	−3.88	O-acyltransferase (WSD1-like) family protein
*Araip.Z2NTP*	*Arahy.D8KQ2T*	4.91	−1.86	0.36	−4.05	−1.73	−1.61	PATATIN-like protein 5
*Aradu.5S6RT*	*Arahy.UL6NUJ*	−0.16	0.57	1.43	0.88	1.04	1.13	1-acyl-sn-glycerol−3-phosphate acyltransferase-like protein
*Araip.4I8QR*	*Arahy.S0JXUX*	−1.56	−0.12	0.88	0.59	0.48	0.65	biotin carboxyl carrier acetyl-CoA carboxylase
*Araip.QK1BG*	*Arahy.Y1617F*	−1.17	−0.32	−1.44	−1.18	−1.1	−1.27	phospholipase D P2
*Aradu.82M2C*	*Arahy.CC9CW3*	0.82	−1.03	2.01	1.27	1.1	1.45	hydroxymethylglutaryl-CoA synthase-like
*Araip.95KUY*	*Arahy.W7LA7X*	−0.12	−0.57	−1.69	−1.95	−1.34	−1.64	patatin-like protein 6
*Aradu.J6P55*	*Arahy.IFJ1V3*	−1.6	−0.92	1.44	−2.31	−0.71	−0.62	3-ketoacyl-CoA synthase 11
*Aradu.4R68R*	*Arahy.IBW8RK*	−0.31	0.31	2.42	−0.21	0.87	1.09	sterol C4-methyl oxidase 1-2
*Aradu.Y6Q3B*	*Arahy.H1181I*	−1.98	−0.41	0.13	−1.07	−0.37	−0.36	fatty acid hydroxylase superfamily
*Aradu.8Y5FN*	*Arahy.TAF7VU*	−0.32	0.62	2.5	0.79	1.25	1.4	niemann-pick C1-like protein
*Aradu.32LC7*	*Arahy.HNK90J*	−0.38	−1.75	−2.19	−3.58	−2.33	−2.64	myo-inositol-1-phosphate synthase 3
*Aradu.07VYH*	*Arahy.JC4ZCG*	−0.98	0.04	4.75	−0.81	1.26	1.81	3-hydroxyacyl-[acyl-carrier-protein] dehydratase FabZ
*Araip.VC3BC*	*Arahy.6JMM3R*	−1.72	−3.33	−0.17	−1.86	−1.83	−1.47	lecithin:cholesterol acyltransferase family protein
*Aradu.AV1HQ*	*Arahy.722ASC*	−2.44	−1.91	−2.5	−2.13	−2.18	−2.26	fatty acid desaturase 8
*Aradu.PR2BP*	*Arahy.W5G5TD*	0.1	−0.73	−2.35	−1.89	−1.78	−2.04	fatty acid/sphingolipid desaturase, delta-6 desaturase
*Aradu.V9VEN*	*Arahy.HXV4VT*	−1.44	0.08	−1.54	−2.16	−0.87	−1.46	phospholipid-transporting ATPase 9-like isoform X1
*Aradu.IJ0H9*	*Arahy.D341M5*	−2	−0.69	−0.1	0.24	−0.05	0.05	probable LRR (leucine-rich repeat) receptor-like serine/threonine-protein kinase
*Araip.V7FVT*	*Arahy.YVU30U*	−0.02	−0.52	1.8	0.36	0.63	0.95	phosphatidylinositol−4-phosphate 5-kinase 1
*Araip.81A8L*	*Arahy.PKW5X9*	1.95	−1.24	−0.76	−2.52	−1.48	−1.65	monogalactosyldiacylglycerol synthase 2
*Araip.T755Q*	*Arahy.NH00NZ*	0.36	−3.44	−2.36	−2.09	−2.43	−2.31	type I inositol 1,4,5-trisphosphate 5-phosphatase 11-like
*Aradu.TI2CX*	*Arahy.6XGZ20*	1.03	0.31	2.11	−1.83	0.45	0.5	alcohol dehydrogenase
*Aradu.AMA0U*	*Arahy.IGY6G6*	−0.49	−0.28	0.86	−0.55	0.08	0.17	LAG1 longevity assurance homolog 3, LAC1
*Araip.0E4ZE*	*Arahy.LFL6K6*	−0.83	−1.04	−1.5	−0.94	−1.21	−1.24	non-specific lipid-transfer protein
*Aradu.CXJ5P*	*Arahy.XQ2AST*	−2.21	−1.55	−0.69	−2.23	−1.56	−1.56	solanesyl diphosphate synthase 1
*Aradu.03ENG*	*Arahy.L0R0IL*	−0.41	−1.52	−1.76	−1.82	−1.68	−1.75	non-specific lipid-transfer protein
*Aradu.0L9GE*	*Arahy.BI8WMI*	0.35	−0.85	−2.34	−2.68	−1.74	−2.2	glycerol−3-phosphate acyltransferase
*Araip.79E4D*	*Arahy.Z1TE7K*	0.21	−1.23	−1.17	0.18	−0.7	−0.63	non-specific phospholipase C3
*Aradu.9G3ZE*	*Arahy.LLG5IK*	1.45	−0.55	−2.76	−0.34	−1.38	−1.51	alcohol dehydrogenase 1
*Araip.3B5FU*	*Arahy.R19W3K*	−0.04	0.02	−0.67	0.52	−0.13	−0.16	lycopene cyclase, lycopene cyclase-type, FAD (flavin adenine dinucleotide) -binding
*Araip.VA90H*	*Arahy.Z9XA8H*	−1.65	−0.34	1.35	0.15	0.4	0.64	acyl carrier protein 4
*Araip.AT3RC*	*Arahy.68T27Q*	−0.8	−0.35	−1.65	−0.55	−1.02	−1.13	Protein kinase superfamily protein
*Araip.N4RJ5*	*Arahy.68T27Q*	−0.97	−0.02	−1.96	−1.45	−0.91	−1.38	putative phospholipid-transporting ATPase 9-like isoform X1
*Araip.L7Z6B*	*Arahy.PUH2BT*	−0.91	−0.39	−0.66	−0.1	−0.38	−0.38	glycerophosphodiester phosphodiesterase GDE1
*Araip.P3CAI*	*Arahy.JGYC2J*	−2.17	−0.27	−1	−0.08	−0.49	−0.53	GDSL-like lipase/acylhydrolase superfamily protein
*Araip.TF5RD*	*Arahy.PY4INI*	−0.42	0.93	2.05	0.31	1.09	1.12	non-specific phospholipase C6
*Aradu.XM2MR*	*Arahy.E24IL6*	1.53	2.06	−1.42	−1.1	−3.71	−2.54	acyl-[acyl-carrier-protein] desaturase, *SAD2* homologous gene
*Araip.3S8HT*	*Arahy.05783X*	0.18	−0.51	−0.06	−0.34	−0.32	−0.24	3-oxoacyl-[acyl-carrier-protein] synthase 3 FabH
*Araip.CW7EC*	*Arahy.JEY2CC*	1.42	−0.48	−0.77	−2.39	−0.93	−1.27	polyketide cyclase/dehydrase and lipid transport superfamily protein
*Aradu.CH0GR*	*Arahy.74RI4B*	1.71	−0.51	−0.68	0.56	−0.22	−0.2	GDSL-like lipase/Acylhydrolase superfamily protein
*Aradu.YH03P*	*Arahy.6FI8AN*	6.52	−0.73	−3.05	−1.9	−2.47	−2.6	alpha/beta-hydrolases superfamily protein
*Araip.QM8AJ*	*Arahy.K79JM6*	−2.79	−1.96	−1.8	−4.75	−2.28	−2.51	glycerol−3-phosphate acyltransferase 2
*Araip.8V7F0*	*Arahy.D341M5*	−1.45	0.25	0.77	−0.12	0.27	0.28	probable LRR receptor-like serine/threonine-protein kinase X3
*Araip.Q3H1S*	*Arahy.FH5Z1U*	0.31	−3.42	−0.43	−2.38	−1.94	−1.47	lipid transfer protein
*Aradu.13SFN*	*Arahy.ZE5M4X*	1.18	0.57	1.49	2.47	1.8	1.91	GDSL-like lipase/acylhydrolase superfamily protein
*Araip.Q7M7G*	*Arahy.BI8WMI*	0.41	−0.83	−2.16	−2.81	−1.68	−2.14	glycerol−3-phosphate acyltransferase 2C chloroplastic-like isoform X2
*Araip.R99KE*	*Arahy.6H2079*	0.39	1.42	0.87	1.18	1.13	1.05	uncharacterized protein LOC100792273 isoform X2
*Aradu.B4FEF*	*Arahy.TZMQ1W*	1.54	−0.54	−1.37	−1.84	−1.04	−1.37	Polyketide cyclase/dehydrase and lipid transport superfamily protein
*Aradu.8019V*	*Arahy.6WS8RX*	5.83	0.22	−2.99	−0.68	−1.86	−1.95	GDSL-like lipase/acylhydrolase
*Aradu.L86SE*	*Arahy.PKW5X9*	2.29	−1.4	−0.92	−2.62	−1.64	−1.81	monogalactosyldiacylglycerol synthase 2
*Araip.S3GXY*	*Arahy.5913QL*	−1.88	−1.39	0.06	−2.55	−0.78	−0.65	fatty acid desaturase 2
*Aradu.UL8XP*	*Arahy.Q6L6L9*	−1.09	−1.36	−1.61	−0.54	−1.19	−1.17	3-hydroxy−3-methylglutaryl-coenzyme A reductase-like protein
*Araip.5UJ5P*	*Arahy.3WFA2L*	1.94	0.29	0.04	0.55	0.32	0.33	long-chain acyl-CoA synthetase 2
*Aradu.TA7RE*	*Arahy.F8M8GF*	6.23	7.03	6.94	7.58	7.22	7.26	Pleckstrin homology (PH) domain-containing protein
*Araip.31Q5V*	*Arahy.GU7MJ6*	−2.11	−1.23	−3.16	−0.87	−1.68	−1.81	fatty acid desaturase 5, plant stearoyl-acyl-carrier protein
*Araip.K56RN*	*Arahy.NDAY03*	0.61	−0.79	−0.54	−0.81	−0.72	−0.71	seed linoleate 9S-lipoxygenase
*Aradu.H0G14*	*Arahy.5A3JDR*	−2.32	1.93	−2.92	−1.34	−0.98	−1.66	12-oxophytodienoate reductase 1
*Aradu.AS232*	*Arahy.7CJ2RM*	0.07	−1.84	−2.3	−1.31	−1.87	−1.87	lipoxygenase 3
*Araip.220UP*	*Arahy.K697CU*	−2.13	−0.02	−0.4	0.42	−0.02	−0.02	acyl-CoA-binding domain 3

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
