# Peer review of "Transcriptomic Analysis Reveals the High-Oleic Acid Feedback Regulating the Homologous Gene Expression of Stearoyl-ACP Desaturase 2 (SAD2) in Peanuts"

_ijms, 2019, doi:10.3390/ijms20123091_

Reviewer 1 Report

The manuscript reports data on the transcriptional characterization of developing seeds of two peanut genotypes differing for oleic acid content. They provide evidence that the high oleic genotype (H176) is due to a mutation in the FAD2 gene. The authors then analysed differentially expressed genes (DEGs) involved in lipid metabolism and suggest a role of the redox signaling system in the feedback regulation of the stearoyl-ACP deasutrase 2 gene (SAD2) in the H176 genotype containing a FAD2 defective gene.

Overall the reported study is properly done, and the results are interesting, however the manuscript needs major revisions in the presentation of the data and together with a strong improvement of the English language. Furthermore the discussion is very speculative as poor data are provided to support the proposed mechanisms of regulation

Please find below some of my major concerns:

Introduction 

Line 96: I do not agree that the authors provide valuable genetic resources, their transcriptomic data and findings provide knowledge for elucidating the mechanisms involved in OA biosynthesis and regulation in peanut.

Materials and Methods

Please describe better the origin of the plant materials: the authors refer to the line H176 as a high-oleic mutant line, does this line originate from a mutagenesis program? If not, it means it is a spontaneous mutation: please clarify

Lines 106-107: rephrase

Line 169: Patel et al 2004, insert number of the reference

Results

Lines 211-212: the content of total oil is not so different between the two cultivars, but this is not the case for OA and LOA, which content is significantly different at all development time!

Lines 224-226: it would be better introducing the cloning of the FAD2 gene explaining why they got two genes. Furthermore, as they introduce FAD genes telling that in peanut there are 6 candidate isoforms (line 67), which criteria did they used to isolate the FAD2 gene?

Line 249: please provide a description and reference of the genome-guided strategy Trinity used.

Line 248: what means the “last” transcriptome?

Line 265-266: DEGs are reported for being identified at six stages in table S4, however in the table only one data is reported for each DEG; to which time point the data is referred?

Line 283: is it correct the reference to table S5?

Line 297: is it correct the reference to table 2B and figure S2?

Lines 301-302: what stated is not shown in figure S5

Line 304: is it correct the reference to figure S7?

Lines 305-306: DEGs instead of DEPs, please indicate which are the mentioned 6 DEGs specifically mapped to unsaturated FA biosynthesis

Line 335: DEG instead of DEL

Lines 336-342: Please rephrase as it is not clear

Line 372: which is the mentioned domain of the FAD? Please show it

Figure 7: panel A, please modify Metaphase with middle stage

Discussion

Figure S7: please insert a reference bar for the level of expression associated to the color

Line 494: why fortunately?

Author Response

Comments and Suggestions for Authors

The manuscript reports data on the transcriptional characterization of developing seeds of two peanut genotypes differing for oleic acid content. They provide evidence that the high oleic genotype (H176) is due to a mutation in the FAD2 gene. The authors then analysed differentially expressed genes (DEGs) involved in lipid metabolism and suggest a role of the redox signaling system in the feedback regulation of the stearoyl-ACP deasutrase 2 gene (SAD2) in the H176 genotype containing a FAD2 defective gene.

Overall the reported study is properly done, and the results are interesting, however the manuscript needs major revisions in the presentation of the data and together with a strong improvement of the English language. Furthermore the discussion is very speculative as poor data are provided to support the proposed mechanisms of regulation 

Please find below some of my major concerns:

1. Introduction 

Line 96: I do not agree that the authors provide valuable genetic resources, their transcriptomic data and findings provide knowledge for elucidating the mechanisms involved in OA biosynthesis and regulation in peanut.

Response: Thank you for your comment. We have revised this sentence according to your advice.

Line 92-94: Our findings provide key knowledge on the mechanisms involved in OA biosynthesis and regulation in high-oleic peanut.

2. Materials and Methods

Please describe better the origin of the plant materials: the authors refer to the line H176 as a high-oleic mutant line, does this line originate from a mutagenesis program? If not, it means it is a spontaneous mutation: please clarify

Response: Actually, H176 (Kainong176) and L70 (Kainong70) are derived from the interbreeding of Kainong30 with Kaixuan01-6. H176 is not originated from a mutagenesis program. The reason of H176 with high-oleic content is attributed to the inheritance of the mutant FAD2-A and FAD2-B from the high-oleic parent Kaixuan01-6. The detailed breeding process of H176 has been described in Figure S2(Line 101). 

Lines 106-107: rephrase

Response: We have revised this part. We mean that H176 and L70 were provided by Kaifeng Academy of Agriculture and Forestry, and we re-cultivated them in the field of Baiyun District of Guangzhou City.

Line 101-103: These two cultivars were provided by Kaifeng Academy of Agriculture and Forestry, and re-cultivated to flowering in the field of Baiyun District of Guangzhou City (Guangdong Province, China) under the normal growth conditions. 

Line 169: Patel et al 2004, insert number of the reference

Response: We have inserted the correct reference number according to the reference format of IJMS.

Line 166: reference number [10], Patel, M.; Jung, S.; Moore, K.; Powell, G.; Ainsworth, C.; Abbott, A., High-oleate peanut mutants result from a MITE insertion into the FAD2 gene. Theor Appl Genet, 2004, 108, (8), 1492-502.

 3. Results

Lines 211-212: the content of total oil is not so different between the two cultivars, but this is not the case for OA and LOA, which content is significantly different at all development time!

Response: Actually, we firstly investigated the content of total oil in the normal- and high-oleic peanut seeds, because we wanted to determine whether the different OA content between the two cultivars produced a negative effective on oil accumulation. Based on the result of total oil content, we concluded that OA accumulation did not affect the total oil content. According to our statistical results, the contents of OA and LOA presented obvious difference at all seed development stages in H176 comparing with that in L70. 

Lines 224-226: it would be better introducing the cloning of the FAD2 gene explaining why they got two genes. Furthermore, as they introduce FAD genes telling that in peanut there are 6 candidate isoforms (line 67), which criteria did they used to isolate the FAD2 gene?

Response: For the first question, we have revised this part as Line 223-225: cultivated peanut is a tetraploid species, whose genome consists of two diploid subgenomes A. duranensis (AA) and A. ipaensis (BB). Therefore, AhFAD2 contains two homologous genes (AhFAD2-A and AhFAD2-B) with similar function.

For the second question, Wang et al. (2105) reported that 6 candidate isoforms of FAD2 were isolated in peanut genome, including microsomal ω-6 fatty acid desaturase (AhFAD2-1, -2, -3, -4) and the chloroplast ω-6 fatty acid desaturase (AhFAD2-5, and -6). Their article suggested that six novel full-length cDNA sequences (named as AhFAD2-1, -2, -3, -4, -5, and -6) were identified in peanut (Arachis hypogaea L.). An analysis revealed open reading frames of 379, 383, 394, or 442 amino acids. We thought that the AhFAD2-1 (379 amino acids) was consistent with we cloned FAD2 (1140bp, 379 amino acids) in normal peanut (L70), and they determined the AhFAD2-1 gene played a major role in the conversion of oleic to linoleic acid during seed development. However, we do not know their isolation criteria, when they published this article, the peanut genome sequence not been released. From our experience, we speculated that the homology cloning strategy is used by referencing the arabidopsis or soybean genome to design primer and amplifying the six FAD2s from the cDNA library. 

Line 249: please provide a description and reference of the genome-guided strategy Trinity used.

Response: As suggested, we have provided a description and reference of Trinity. Line 145-147: RNA-library sequencing and assembly were performed by Novogene Ltd. based on the Trinity [26] method, the processes of Trinity method consisted of three software modules: Inchworm, Chrysalis, and Butterfly, applied sequentially to process large volumes of RNA-Seq reads.

Line 248: reference number [28], Grabherr, M.G.; Haas, B.J.; Yassour, M.; Levin, J.Z.; Thompson, D.A.; Amit, I.; Adiconis, X.; Fan, L.; Raychowdhury, R.; Zeng, Q.; Chen, Z.; Mauceli, E.; Hacohen, N.; Gnirke, A.; Rhind, N.; Di Palma, F.; Birren, B.W.; Nusbaum, C.; Lindblad-Toh, K.; Friedman, N.; Regev, A., Full-length transcriptome assembly from RNA-Seq data without a reference genome. Nature Biotechnology, 2011, 29(7): 644–652. 

Line 248: what means the “last” transcriptome?

Response: We have removed the “last”, we mean that all of last assembled transcriptome library were used for further analysis. Line 249-250. 

Line 265-266: DEGs are reported for being identified at six stages in table S4, however in the table only one data is reported for each DEG; to which time point the data is referred?

Response: All the DEGs data are merged into Table S4. We have separated the DEGs at each time point in different sheet, and the new Table S4 clearly displays these data. 

Line 283: is it correct the reference to table S5?

Response: It is correct, we have revised the reference of table S5 with separate sheets. Line 287.

 Line 297: is it correct the reference to table 2B and figure S2?

Response: We have revised the reference of table 2B and figure S2. Original figure S2 has been replaced by figure S4. Line 259. 

Lines 301-302: what stated is not shown in figure S5

Response: I am very sorry for our mistake, we have revised the part of “3.4 Expression analysis of DEGs” (Line 299), original figure S5 has been replaced by figure S7 (Line 305-309). 

Line 304: is it correct the reference to figure S7?

Response: The original figure S7 has been replaced with a new figure S10, and we have revised the reference of figure S10 (Line 456)..

Lines 305-306: DEGs instead of DEPs, please indicate which are the mentioned 6 DEGs specifically mapped to unsaturated FA biosynthesis.

Response: Thank you very much, we have corrected this error. We have provided the gene names that were mapped to unsaturated FA biosynthesis, and a new figure S10 has been provided, and the gene names were displayed in Table S12. 

Line 335: DEG instead of DEL

Response: We have corrected this error. 

Lines 336-342: Please rephrase as it is not clear.

Response: We have revised this part. Line 341-346. 

Line 372: which is the mentioned domain of the FAD? Please show it

Response: FAD2 coded protein contains 379 amino acids and two conserved domains, including the DUF3474 (5-63aa) and FA-desaturase (78-345aa) domains. The presence of FA-desaturase domain suggests that FAD2 is able to target endoplasmic reticulum. The FAD2 protein structure is presented in the new Figure S9 (Line 378).

 Figure 7: panel A, please modify Metaphase with middle stage

Response: Thank you very much, we have provided a new Figure 7 without “metaphase”.  

4. Discussion

Figure S7: please insert a reference bar for the level of expression associated to the color.

Response: The original figure S7 has been replaced with a new figure S10, and the reference bar has been inserted into the new figure S10. 

Line 494: why fortunately?

Response: The fortunately has been replaced with “In addition” (Line 494).   

As you can see, we have tried our best to address all the concerns from the reviewers. We hope it is now suitable for publication. If you have any questions concerning the manuscript, please contact us.  We are Looking forward to hearing from you. Thank you and best regards.

Reviewer 2 Report

The study has generated extensive transcriptomic data but the MS is not justifying the efforts. Authors need significant efforts to improve scientific writing. 

Abstract

 Line 17 “comparative transcriptome was conducted” –should be – “comparative transcriptomics was conducted”

“First, we identified the mutant type of FAD2 that is capable of utilizing a potential germplasm resource for future high-oleic peanut breeding” – not clear, do you mean identified allele? 

“Further transcriptome characterized 74 differentially expressed genes (DEGs) involved..” – need significant improvement. It should be – Transcriptomic analysis identified 74 DEGs. 

“five DELs encoded” – typo –“ five DEGs encoded”

“To examined the dynamic mechanism of this finding” – should be “To examine the dynamic mechanism of this finding” 

Introduction

“productivity of more than 40 million tons in global yield” – this is production and not a productivity

There are plenty of mistakes throughout the MS, I hope Authors will rectify those in the revised version. 

Methods 

“Peanut genome size is approximately 2.7 GB, but the complete nucleotide sequence is not released, because the close genetic relationship between two diploid species is technically difficult to assemble the genome” – not true, the genome sequence for cultivated Peanut is published. Again, there is plenty of transcriptome data is also available. 

“ library was mixed by three individual peanut seed sample” – Biological replications need to mention here. 

Cut-off - Log2 (Fold Change) value of >1 and <-1, respectively, as well as the P < 0.05 is not appropriate particularly when you have six biological replications. I suggest to use >2 and <-2 and Bonferroni corrected p-value cutoff.  

Results

Authors need to have better pathway enrichment analysis with the DEGs. To confirm the association of DEGs with the OA, need to perform qPCR in 4-5 lines with different genetic background. 

Author Response

Comments and Suggestions for Authors

The study has generated extensive transcriptomic data but the MS is not justifying the efforts. Authors need significant efforts to improve scientific writing. 

1. Abstract

Line 17 “comparative transcriptome was conducted” –should be – “comparative transcriptomics was conducted”

Response: Thank you very much, we have revised this sentence (Line 18).

“First, we identified the mutant type of FAD2 that is capable of utilizing a potential germplasm resource for future high-oleic peanut breeding” – not clear, do you mean identified allele?

Response: The FAD2 we identified was not a new allele of FAD2 locus, we just determined the mutant type of FAD2 in high-oleic cultivar H176 (Kainong176), and we mean that the H176 was used as a germplasm resource for future high-oleic peanut breeding (Line 20-23).

“Further transcriptome characterized 74 differentially expressed genes (DEGs) involved..” – need significant improvement. It should be – Transcriptomic analysis identified 74 DEGs.

Response: we have revised this sentence (Line 23-25).

“five DELs encoded” – typo –“ five DEGs encoded”

Response: I am sorry for the mistake, we have corrected this error.

“To examined the dynamic mechanism of this finding” – should be “To examine the dynamic mechanism of this finding”

Response: Thank you very much, we have corrected this error (Line 28).  

2. Introduction

“productivity of more than 40 million tons in global yield” – this is production and not a productivity

Response: Thank you very much, we have revised this sentence according to your advice (Line 42). 

There are plenty of mistakes throughout the MS, I hope Authors will rectify those in the revised version.

Response: Thank you very much, the revised manuscript has been edited by a native English speaker.

3. Methods

“Peanut genome size is approximately 2.7 GB, but the complete nucleotide sequence is not released, because the close genetic relationship between two diploid species is technically difficult to assemble the genome” – not true, the genome sequence for cultivated Peanut is published. Again, there is plenty of transcriptome data is also available.

Response: our team recently published the cultivated peanut genome (Fuhuasheng): Sequencing of Cultivated Peanut, Arachis hypogaea, Yields Insights into Genome Evolution and Oil Improvement (DOI:https://doi.org/10.1016/j.molp.2019.03.005), and two articles also published the cultivated peanut Shitouqi and Tifrunner genome sequence on May 01, 2019. But when we began this project of high-oleic peanut transcriptome sequencing, only the diploid species genome could be referenced for assembling the transcriptome library. Now, we have revised this part in the revised version, and the reference numbers have been inserted into the Line 73. 

“ library was mixed by three individual peanut seed sample” – Biological replications need to mention here.

Response: we have revised this part (Line 136).

Cut-off - Log2 (Fold Change) value of >1 and <-1, respectively, as well as the P < 0.05 is not appropriate particularly when you have six biological replications. I suggest to use >2 and <-2 and Bonferroni corrected p-value cutoff.

Response: previously, we want to obtain more DEGs at different time-points; therefore we adopted the Cut-off-Log2 (Fold Change) value of >1 and <-1. we="" now="" agree="" with="" your="" and="" previous="" submitted="" degs="" at="" different="" stages="" the="" cut-off-log2="" fold="" value="" of="">1.3 and <-1.3 (Table S4). In addition, I am very appreciate for your advice that encourages us to improve the screening criteria of DEGs in the next study. 

4. Results

Authors need to have better pathway enrichment analysis with the DEGs.

Response: Thank you very much for your advice. Actually, original enrichment analysis provided by the Dunzhou Mo (Capitalbio Corporation), and then we re-plotted the result with figure S7B according to the table S6. Here, we mainly focus on the fatty acid biosynthesis and metabolism pathway, and 74 DEGs were identified at different stage of peanut seed development.

To confirm the association of DEGs with the OA, need to perform qPCR in 4-5 lines with different genetic background.

Response: Transcriptomics analysis provided many DEGs between the two different cultivars, and we focused on the fatty acid desaturase family of genes. Therefore, their expressions in H176 and L70 were investigated. To date, we are collecting the high-oleic variety with different genetic background, but we have to identify the mutant type of FAD2 in those collected high-oleic peanut variety. We did not have the qPCR data now, and your advice encourages us to examine the relative expression levels of DEGs involved in OA synthesis in the next study. 

As you can see, we have tried our best to address all the concerns from the reviewers. We hope it is now suitable for publication. If you have any questions concerning the manuscript, please contact us.  We are Looking forward to hearing from you. Thank you and best regards.

Round  2

Reviewer 1 Report

The new revised version of the manuscript has consistently improved, however it still remains necessary an english revision as some parts of the manuscript are low quality.

Author Response

The new revised version of the manuscript has consistently improved, however it still remains necessary an english revision as some parts of the manuscript are low quality. 

Response: We really appreciate your comment, and the re-submitted manuscript has been edited by a native English speaker again. We have revised some parts with low English quality in the track changes version. We believe that our modified version is in accord with the requirements and standards for publishing in IJMS. 

Reviewer 2 Report

I still suggest going for transcriptome analysis with the released genome draft an updated version because the genome is highly complex and duplicated.  Does not matter how many articles earlier published without genome, but now it is available and must be used efficiently.

Author Response

2. Comments and Suggestions for Authors

I still suggest going for transcriptome analysis with the released genome draft an updated version because the genome is highly complex and duplicated. Does not matter how many articles earlier published without genome, but now it is available and must be used efficiently. 

Response: We really appreciate your suggestion. Previous raw data of our transcriptome library was analyzed by Capitalbio Corporation (Dunzhou Mo, Beijing), which produced gene accession numbers mainly referencing the diploid ancestor’s genome. When the cultivated peanut draft genome was released, the bioinformatics reanalysis project was entered into our upcoming schedule, however this work really requires significant time and we will provide this result in the next study. Herein, the 74 identified differentially expressed lipid genes are the core content in this article, and therefore their accession numbers (Araip.XXXX or Aradu.XXXX) have been transformed with new format (Arahy.XXXX in Table 2) by referencing the draft genome of cultivated peanut.

Despite the release of the draft genome, its accuracy should be evaluated by the actual experiments. We did not clone multiple fatty acid genes’ promotor sequences according to the draft genome to design primers, including FAD8 and SAD2. However, we were able to clone the interested gene from the cultivated peanut’s cDNA library by directly using the designed primer based on the diploid ancestor’s transcriptome sequence (FAD8, KAS2, ASN1, FAB2, FAD2, AHL10, Sus1, GA3oxase). Our result implies that the released transcriptome sequences of the diploid ancestor and cultivated peanut are very similar. The two diploid progenitors’ sequence information are available along with predicted genes and descriptions. The genomes of the diploid progenitors have been used to help identify and assemble similar chromosomes in the cultivated peanut. Therefore we believe that referencing the diploid genome in the analysis of the cultivated peanut transcriptomic is appropriate.

 Additionally, our transcriptomic study provides a potential resource for understanding the variation tendency of gene expression in the peanut seed, and the most important function of this transcriptomic dataset is we can use the targeted cloning of differentially expressed genes (DEGs, Table S3) for follow-up study. For example, once the researcher intends to clone their interested DEGs, the gene accession number (Araip.XXXX or Aradu.XXXX) based on reference the genome of diploid ancestor could be submitted to the gene name blast module of “Gene & Gene family” in the Peanutbase homepage (www.peanutbase.org). The obtained CDS sequence of interested DEG can be used to blast the genome of cultivated peanut, and therefore the original accession number (Araip.XXXX or Aradu.XXXX) is reflected on the cultivated peanut genome (Arahy.XXXX). Subsequently we can design the primers to clone the full length CDS from the cultivated peanut cDNA library utilizing the new accession number (Arahy.XXXX). 

The detail process has been described in the attachment  file(PDF version).

Round  3

Reviewer 2 Report

I can accept the explaination provided by the Authors about the use of updated draft of genome  seqquence to perform RNAseq analysis.